# Is it feasible to implement a community-based participatory group programme to address issues of access to healthcare for people with disabilities in Luuka district Uganda? A study protocol for a mixed-methods pilot study

Hannah Kuper ,[1] Andrew Sentoogo Ssemata,[2] Tracey Smythe ,[1,3]
Joanna Drazdzewska,[4] Peter Waiswa,[5,6] Patrick Kagurusi,[7] Mikey Rosato,[4]
Femke Bannink Mbazzi[1,2]

For numbered affiliations see end of article.

**Correspondence to**
Dr Hannah Kuper;
hannah.kuper@lshtm.ac.uk

## ABSTRACT

**Introduction** On average, people with disabilities face many difficulties in accessing healthcare and experience worse health outcomes. Yet, evidence on how to overcome these barriers is lacking. Participatory approaches are gaining prominence as they can generate low-cost, appropriate and scalable solutions. This study protocol is for the pilot testing of the co-created Participatory Learning and Action for Disability (PLA-D) groups to assess feasibility.

**Methods and analysis** We will pilot test PLA-D in five groups in Luuka district, Uganda during 2023. Each group will include approximately 20 members (people with disabilities, family members, carers) who will meet every 2–3 weeks over a 9–11 month period. The groups, guided by a trained facilitator, will identify issues about health and healthcare access and plan and implement locally generated solutions (eg, raising awareness of rights, advocacy and lobbying, establishing health savings and financing schemes). We will collect diverse sources of data to assess feasibility: (1) in-depth interviews and focus group discussions with group participants, non-participants and group facilitators; (2) monitoring of group activities; (3) direct observation of groups and (4) quantitative survey of group participants at baseline and endline. Data analyses will be undertaken to assess feasibility in terms of: acceptability, demand, implementation and practicality. We will develop and refine evaluation tools in preparation for a future trial.

**Ethics and dissemination** Ethical approval for the study has been received by the London School of Hygiene & Tropical Medicine and the Uganda Virus Research Institute ethics committees. Informed consent will be obtained from all study participants, making adaptations for people with disabilities as necessary. We will reach different groups for our dissemination activities, including (1) people with disabilities (eg, community meetings); (2) policy and programme stakeholders in Uganda and international (eg, individual meetings, evidence briefs) and (3) academics (journal articles, conference/seminar presentations).

## STRENGTHS AND LIMITATIONS OF THIS STUDY

⇒ Key strength of the pilot study is that the feasibility of the programme will be assessed using mixed-method approaches, with participation from people with disabilities.
⇒ A limitation is that the pilot study is only in one setting and will therefore allow assessment of some domains of the Bowen feasibility framework (acceptability, demand, implementation, practicality) but not others (adaptation, integration, expansion, limited efficacy).
⇒ The limited number of participants included in the pilot study may make it difficult to draw inferences for feasibility for different subgroups, such as people with different impairment types.

## INTRODUCTION

There are at least 1.3 billion people with disabilities globally, more than 80% of whom live in low and middle-income countries (LMICs).[1 2] People with disabilities include those who have long-term physical, mental, intellectual or sensory impairments which, in interaction with various barriers, may hinder their full and effective participation in society on an equal basis with others.[2 3] People with disabilities face a wide range of discrimination and negative attitudes.[2] They are consequently more likely to be poor, and excluded from education, employment and societal participation.[2] Another key challenge facing many people with disabilities is exclusion from healthcare and difficulties achieving good health,[1 2 4–7] which is the focus of the current study.

People with disabilities frequntely experience worse health than others in the population.[1 2 4–6] By definition, people with disabilities have an underlying health condition and impairment (eg, diabetes, physical impairment), which are linked to other health risks (eg, stroke, pressure sores). They are also on average older, poorer and more marginalised, and have a higher prevalence of a range of risk factors (eg, violence, physical inactivity, diabetes, hypertension).[1 2 6] People with disabilities therefore, on average, have greater general healthcare needs than others in the population.[1 4] They will also require regular healthcare services like anyone else in the population (eg, vaccinations) and potentially specialised healthcare services (eg, physiotherapy). However, they frequently experience widespread barriers to accessing healthcare, including lack of accessible transport and facilities, poor skills of healthcare providers around disability and high costs.[1 7] Consequently, people with disabilities are 41%–57% more likely to have unmet healthcare needs, according to the WHO's World Report on Disability.[2] Quality and affordability of healthcare services are also often worse for people with disabilities. They are twice as likely to find healthcare providers' skills and facilities inadequate, three times more likely to be denied healthcare and four times more likely to be treated badly in the healthcare system.[2] The WHO report also found that half of people with disabilities cannot afford healthcare, and they are 50% more likely to suffer catastrophic health expenditure.[2] As a result of all these factors, people with disabilities on average have worse health and higher mortality rates than others in the population, including in the African context.[2 8–11] These patterns can also be observed in Uganda, the context for the current study, where the 2014 census estimated that 12.5% of the population have disabilities.[12] Studies have shown that people with disabilities in Uganda are more likely to experience poverty and exclusion, poor health and difficulties accessing services and information.[12–17]

Improving access to healthcare for people with disabilities is therefore an important priority for a number of reasons. We will fail to achieve health targets and development goals without an explicit focus on disability, including Sustainable Development Goal (SDG) 3 on 'Good health for all' and Universal Health Coverage.[18] Other SDGs, including those related to education and employment, will also not be met if people with disabilities continue to have poor health. Exclusion from healthcare is a violation of rights of people with disabilities, as set out in the United Nations Convention on the Rights of Persons with Disabilities and in the laws of most countries, including Uganda.[18 19] Good healthcare also matters to people with disabilities and their families as it affects their ability to survive and enjoy good health. Moreover, improving healthcare access for people with disabilities has the potential to be cost-saving,[1] as it may prevent unnecessary costs for the health system and improve health services for all through universal design.

## Rationale for development of a participatory intervention to improve access to healthcare for people with disabilities

Two systematic reviews on access to general and specialist healthcare for people with disabilities in LMICs found few examples of interventions to address the widespread gaps observed.[20 21] An Evidence and Gap Map on disability-inclusive development also failed to identify good examples of disability-inclusive health interventions.[22] The WHO has made specific recommendations for the promotion of disability-inclusive health,[1 2] including (1) removing physical barriers to health facilities, information and equipment, (2) making healthcare affordable, (3) training all healthcare workers in disability issues including rights and (4) investing in specific services such as rehabilitation. These 'top-down' interventions are important, but unlikely to be sufficient since barriers will vary locally.[7] Participation of people with disabilities is therefore critically important for developing new locally appropriate solutions—particularly in light of the 'Nothing about us, without us' ethos of the disability movement. Participatory approaches are also recommended by the new WHO report on health equity for persons with disabilities.[1]

Participatory approaches are gaining attention from the global health community as they can empower communities to work together to identify local issues and together generate low-cost, appropriate and scalable solutions. A leading example is the Participatory Learning and Action (PLA) approach, which was designed to reduce maternal and neonatal mortality.[23] Through PLA, women's groups are convened and facilitated to identify underlying barriers to care and care-seeking and then develop, implement and evaluate local solutions to address these issues (figure 1). Examples of solutions implemented include the creation of emergency crisis funds to pay for medical care for those in poverty, arranging a bicycle ambulance to bring people to hospital and lobbying local health authorities for more staff. PLA supports care-seeking and home-based care practices through these pathways, thereby improving health and well-being and reducing mortality in women and newborns. A meta-analysis of seven randomised controlled trials (RCTs) from four countries found that PLA was associated with a 22% reduction in maternal mortality and 20% reduction in neonatal mortality.[23] Subgroup analysis of four RCTs where at least 30% of pregnant women participated in groups showed even greater reductions in both maternal mortality (49%) and neonatal mortality (33%).[23] Evidence also shows that the PLA methodology is sustainable,[24] scalable[25] and equitable.[26] It is recommended by WHO as a strategy for improving maternal and newborn health and reducing mortality, particularly in rural settings where mortality rates are high and access to services is low.[27] PLA has been successfully implemented in 15 countries, including in several settings in Uganda.

To date, PLA has not been used explicitly for people with disabilities. However, evidence from Nepal showed that there was no difference in group attendance between

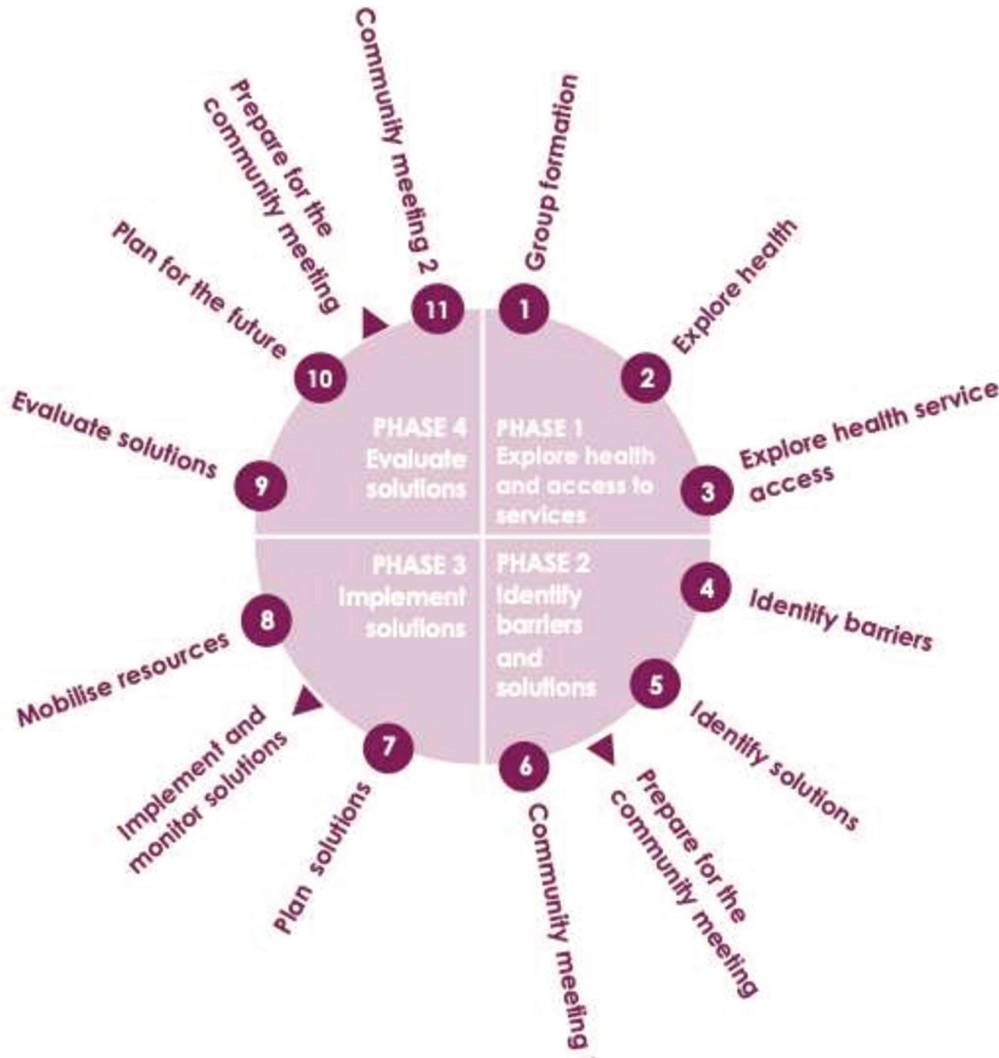

**Figure 1** The PLA-D group cycle has 11 meetings across four phases. PLA-D, Participatory Learning and Action for Disability.

women with and without disabilities.[28] A non-PLA participatory approach has been used in the Philippines to try to improve access to sexual and reproductive healthcare for people with disabilities through establishment of community groups—W-DARE project (Women with Disability taking Action on REproductive and sexual health).[29] Five groups of women with disabilities (established by impairment type) and one group of parents of children with disabilities met fortnightly for 10 meetings to identify and discuss key sexual and reproductive healthcare access issues. A qualitative evaluation highlighted positive benefits of the programme, including increased knowledge and confidence of the women to seek services.[30] Other participatory approaches to promote disability inclusive health are lacking.

We hypothesise that PLA for Disability (PLA-D) would improve healthcare access, health and well-being and reduce the mortality of people with disabilities in Uganda. The PLA-D approach is expected to work from the bottom-up as people with disabilities identify and tackle locally relevant barriers. The potential drivers of action are that group meetings allow people with disabilities to

meet and identify local issues, and develop and implement solutions that are self-resourced. Existing programmes show that solutions implemented address a range of issues, such as lack of awareness (eg, education on health issues and rights), lack of decision-making authority on health (eg, lobby families to fund healthcare of person with disabilities) or poverty (eg, livelihood programmes). The intervention will be supported by health system strengthening activities to improve healthcare access and quality by improving the knowledge, attitudes and skills of healthcare workers and helping identify and address issues around accessibility of facilities. Furthermore, groups also frequently undertake advocacy activities, for instance, to encourage other members of the community to become involved and take action (eg, support transport to healthcare facilities). Groups can also lobby decision-makers to encourage policy and programme change and raise awareness of disability. Through these actions, people with disabilities are anticipated to experience better access to healthcare, improved health and well-being and ultimately reduced mortality. The current protocol focusses on the group meetings (PLA-D). We

have been guided by the Medical Research Council framework in the development of the PLA-D approach, and proposed feasibility assessment, and in the future plan to undertake an evaluation of the intervention.[31]

## Adaptation of PLA for people with disabilities

In 2022, we undertook qualitative research with people with disabilities with a range of impairment types (n=27), healthcare workers (n=15) and other stakeholders, including carers (n=11), to describe barriers and facilitators to accessing healthcare by people with disabilities in Uganda and assess the demand for PLA-D. We also reviewed the literature to identify good practice for improving access to healthcare for people with disabilities, including existing systematic reviews,[20–22] key source documents[2 6 32 33] and a good practice compendium.[34] This activity was supplemented through interviews with key stakeholders, including PLA experts, Organisations of Persons with Disabilities (OPDs), inclusive health experts.

We held a 4-day design workshop in October 2022 to co-create the adapted PLA approach to be accessible, relevant and appropriate for people with disabilities in Uganda (PLA-D), facilitated by Women and Children First. Participants included the research team (n=7), PLA experts (including Amref and Makerere University, n=5), health system stakeholders (eg, District Health Officers, n=2), district disability focal person from the community development office (n=1), people with disabilities and representatives from National OPDs (n=8). The group worked together to agree how to adapt the logistics of the

intervention delivery (eg, identifying who the facilitators should be, where groups should be convened, addressing accessibility) and identified appropriate local implementation and OPD partners (figure 2). The PLA content was then adapted with reference to recommendations from the PLA evidence base and literature, including the facilitator manual, materials (eg, pictures, cards) and session plans. We updated the Theory of Change for PLA to reflect how the intervention is anticipated to have the expected impact (figure 3).

This protocol is for a study to pilot test the co-created PLA-D groups in order to assess their feasibility, and to inform the design of a future cluster RCT).

## METHODS AND ANALYSIS
### Overview
A mixed-methods pilot study will be undertaken in 2023 to assess the feasibility of the PLA-D groups and inform the future development of a cluster RCT. The research will be undertaken in Luuka district, Eastern Uganda, as a partnership between researchers from the London School of Hygiene & Tropical Medicine, MRC/UVRI & LSHTM Uganda Research Unit ('Uganda Research Unit'), Makerere University and the Non-Governmental Organization (NGO) Women and Children First, UK and Amref Health Africa in Uganda. Luuka district had a population of 238 000 in the 2014 census. It is the setting of ongoing research by Makerere, and so Makerere has

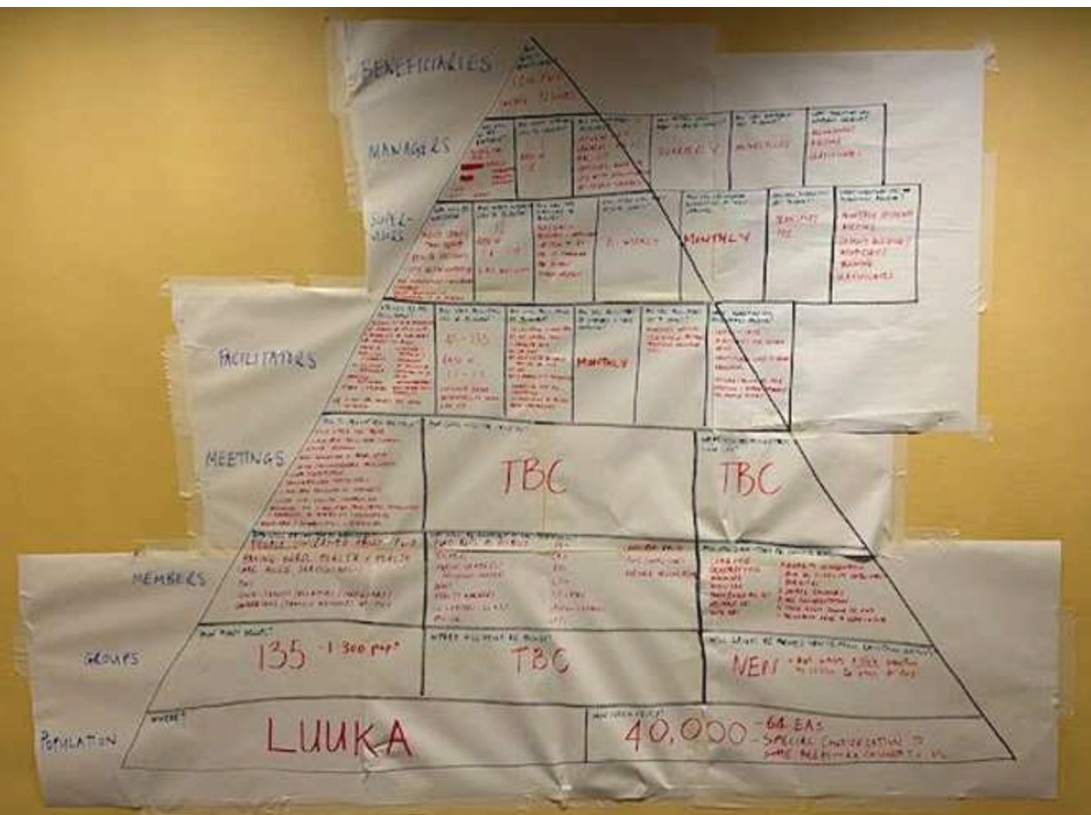

**Figure 2** Decisions reached on design of PLA-D. PLA-D, Participatory Learning and Action for Disability.

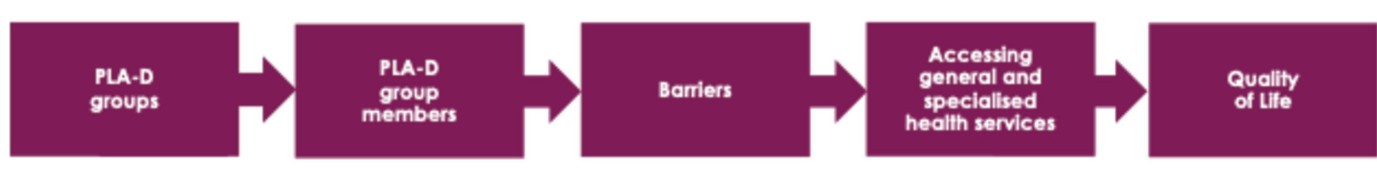

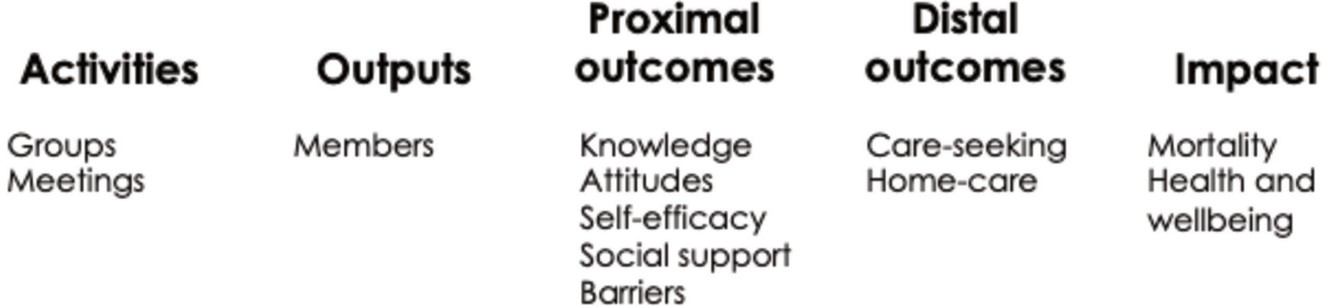

**Figure 3** PLA-D theory of change. PLA-D, Participatory Learning and Action for Disability.

strong existing networks in the district which will facilitate project implementation. The population is >95% rural and almost all households are engaged in agricultural activities.

### Conceptualisation of disability

The UN Convention on the Rights of Persons with Disabilities conceptualises people with disabilities as … 'those who have long-term physical, mental, intellectual or sensory impairments which in interaction with various barriers may hinder their full and effective participation in society on an equal basis with others.'[3] This definition is in line with the interpretation of disability in the Uganda Persons with Disabilities Act, 2020 as 'a substantial functional limitation of a person's daily life activities caused by physical, mental or sensory impairment and environment barriers, resulting in limited participation in society on equal basis with others and includes an impairment specified in Schedule 3 to this Act.' For the purpose of the pilot study, people will self-identify as whether they have a disability or not.

### Pilot of the content and implementation of PLA-D groups

We will establish five pilot PLA-D groups in one sub-country in Luuka district, in May–June 2023. The sample size was selected based on logistical feasibility (time, costs) and our previous experience of pilot testing a group-based intervention suggested that this number was sufficient to assess the feasibility of the intervention.[35]

In collaboration with the District Health Officers, we will select the communities to represent a variety of contexts (eg, very rural or semi-urban, remote or less remote areas). We will work with local community members (eg, community health workers (CHWs) or community health extension workers) to identify people with disabilities in the community. People with disabilities will then be given information about the purpose and structure of the PLA-D groups and invited to attend. Membership will also

be open to family members/caregivers of people with disabilities, although emphasis will be given to the participation of people with disabilities. We anticipate that each group will include approximately 20 people, potentially at least half of whom will be people with disabilities.

Potential group members, with and without disabilities, will be identified and recruited through the community engagement meetings held at the five different parishes (one per group). Information about the study and groups will be provided and those interested in participation will be asked to sign up/register their names and contact and will be contacted individually by the research team. Additional people with disabilities will be invited to attend by the focal person with disabilities at the district level, and by people with disabilities who attended the community engagement meetings.

Groups will meet every 2–3 weeks over a 9-month period (ending December 2023/January 2024), and will go through the PLA cycle (figure 1). The groups will together choose the location of the meetings, ideally selected to promote accessibility, as well as the timing. The PLA cycle includes sessions on identifying issues in accessing healthcare, developing and implementing solutions (eg, savings schemes, growing nutritious food) and self-evaluating their impact. Other community members will be reached through regular community members.

The groups will be facilitated by a facilitator (person with disabilities from the local community and/or a CHW). The facilitators will be supported by trained supervisors and managers, which will be coordinated by the NGO Amref. The facilitators, managers and supervisor will be involved in a week-long hands-on PLA-D training, led by Women and Children First, Amref and the Uganda Research Unit, based on the developed manual. The training will cover aspects including understanding of disability, the PLA-D cycle, how group meetings will be conducted and management and facilitation of the

groups in the community (eg, safeguarding, managing expectations, progress recording and reporting).

## Feasibility assessment

We will use the Bowen framework to consider the feasibility of the PLA-D groups.[36] Our main focus will be on the first four components of the framework: acceptability, demand, implementation and practicality. We will not consider the components of adaptation, integration and expansion for this pilot study, as they relate to issues of scaling existing programmes. Limited efficacy is also not considered, but effectiveness will be considered in a future RCT. We will develop and refine evaluation tools of the groups to inform this future study.

There are four main sources of data collection.

1. Programme monitoring

Attendance of sessions and frequency of sessions will be monitored through a simple register which will be collected by the facilitators.

2. Pre-post quantitative survey (online supplemental materials 1,2)

A survey will be completed by adult PLA-D group participants with disabilities at the programme first and last session (either directly, or via a proxy). These will be developed in English and translated into Luganda and Lusoga (administered in the language of choice of the respondent) and will include:

► Sociodemographic characteristics of the participant.
► Disability assessment (Washington Group Enhanced Short Set).
► Quality of Life (WHO-BREF).
► Health condition and nutrition.
► Healthcare utilisation.
► Participation.
► Attitudes.
► Main goals for the intervention.*
► Review of goals achieved.**
► Satisfaction with group programme.**

   *Only asked at baseline; **only asked at endline.

Questionnaires are based on existing tools (WHOQOL BREF, WOPS questionnaire from cohort in Uganda, WHO Model Disability Survey and SINTEF survey), which have all been widely used. Any potentially controversial questions have been removed (eg, SINTEF questions on violence). A separate questionnaire has been developed for children aged 5–17 years, to be completed by the parent/carer as a proxy. Questionnaire data will not be collected for children <5 years.

All questionnaires will be interviewer-administered in order to maintain consistency and uniformity in interpretation of the questions. Interviews will be undertaken by researchers from the Uganda Research Unit. All questionnaires will be translated in Lusoga and Luganda, the most common local languages in Luuka district. Proxy respondents will be used for adults with difficulties understanding or communicating, and with children aged <18 years. Children aged 8–17 will be asked to give assent to the questionnaire being completed on their behalf.

3. Semistructured interviews and Focus Group Discussions (FGDs) (online supplemental materials 3–6)

Qualitative data will be collected from a range of groups, led by researchers from Uganda Research Unit.

Semistructured in-depth interviews (IDIs) will be conducted with the group facilitators after each session. The researchers will conduct phone interviews with the group facilitator after each session to review the progress of the session, identify areas that went well, challenges that arose and areas for improvement. This information will be used to allow fast-track learning so that future sessions by that or other groups can be adapted in line with recommendations (eg, tips for promoting inclusion, location of meetings, etc).

Additionally, semistructured IDIs and FGDs will be undertaken with group participants and facilitators after the final session. The IDI will be undertaken in the language of choice of the respondent (English or Luganda or Lusoga) with a focus on: (1) satisfaction with the groups, (2) perceived challenges, facilitators and barriers to successful group implementation, (3) group capacity, (4) areas for improvement and (5) perceived positive and negative impacts. Interviews will conducted face-to-face (wherever possible) or remotely. Proxy respondents will be used for adults with difficulties understanding or communicating, and with children aged <18 years.

The IDIs and FGDs will include:

► FGDs with PLA-D participants of the five groups excluding those who participated in the IDIs (6–8 participants per group, 30–40 in total).
► IDIs with 10–15 PLA-D group participants (2–3 participants per group). Participants will be selected to reflect a range of types and severity of disability, gender and socioeconomic profile. They will include people with disabilities and caregivers of persons with disabilities and PLA-D members without disabilities. These interviews will support triangulation of data from the FGDs and further inquiry of areas for further exploration raised during the FDGs.
► IDI with 10–15 people eligible to be PLA-D group participants, but who attended three or fewer sessions (including no sessions). Participants will be selected to reflect a range of types and severity of disability, gender and socioeconomic profile. They will include people with disabilities and caregivers of persons with disabilities.
► IDI with each of the group facilitators (total=5).

Topic guides are available in the supplement.

We anticipate that each interview will take approximately 45–60 min, and the FGD 60–90 min. Carer/proxy interviews will be conducted for people with severe difficulties understanding or communicating even with available adaptations (eg, people with hearing loss, illiterate and with no knowledge of sign language; people with severe intellectual/cognitive impairments). Inclusion of people with disabilities will be supported through sign language interpretation as needed, accessible interview sites and transport, and researchers skilled at communicating with

people with cognitive impairments. Carer/proxy interviews will also be used for children with disabilities below the age of 18 years.

4. Direct observation of groups (online supplemental material 7)

Direct observation of sessions will be undertaken by the local researchers using a checklist guide. The checklist will contain information about the participatory approaches used by facilitators, the level of interest and engagement of participants and notes on any aspects of the group meeting which went well or did not go well.

### Data management and storage

The interviewers will audio record the interviews/FGD. Interviews will be transcribed and translated to English (for those conducted in Luganda or Lusoga). Anonymised transcripts with pseudonyms will be used during the analysis. Quantitative survey data will be anonymised using participant codes and entered into a REDCap database for analysis. All data will be kept as per General Data protection regulation requirements.

Data will be kept strictly confidential (eg, anonymisation of data to remove personal identifiers, upload/download of data through secure networks and password-protecting files). Names of participants will not be given on any documentation and no individuals will be identified in any publication.

### Analysis plan

We will use the data to evaluate the feasibility of PLA-D implementation focusing on four of the Bowen components of feasibility (table 1).[36] Qualitative analysis will be led by two researchers from the Uganda Research Unit (ASS, FBM), who are both Ugandan nationals and both have postgraduate degrees in qualitative research. A thematic approach will be used to analyse qualitative data. The data will be coded using NVivo V.12, specialist software for qualitative data analysis, and the data will be

analysed to develop a fuller framework of themes and subthemes. Particular attention will be given to the feasibility of implementation for people across the full range of impairment types, and whether it addresses the needs of both adults and children.

We will perform a descriptive analysis of quantitative data using STATA by a trained epidemiologist (HK). In particular, we will provide data on the levels of knowledge, attitudes and behaviours, self-rated health and health-seeking behaviour for the study population at baseline and endline. We will analyse data as means with SD, medians with IQRs, or frequencies or percentages, depending on the indicators. Differences in outcomes between baseline and endline will be described (eg, comparing change in means or proportions), using appropriate tests (eg, t-tests, $\chi^2$ tests), comparing both all participants and then participants only with both baseline and endline data. Indicators for inclusion in the future RCT will be considered (eg, based on feasibility of collection, range in answers provided). Indication will also be given of potential effect sizes for the planned RCT, based on pre-post differences, to allow calculation of required sample size.

### Patient and public involvement

The development and pilot testing of PLA-D will be in partnership with people with disabilities. During the development phase this included: sharing plans with OPDs and people with disabilities; interviewing people with disabilities on their perceived healthcare needs and suggestions for group activities; including people with disabilities in the advisory group and in the co-design workshop. People with disabilities will also have strong involvement in the implementation and pilot testing of PLA-D, as: group members; group facilitators (wherever possible); research study participants; researchers/data collectors (wherever possible); advisory group members.

**Table 1** Feasibility assessment of PLA-D: domains explored and data to be used[36]

| Bowen domain | To what extent… | Example outcomes | Data used |
|---|---|---|---|
| Acceptability | … is PLA-D judged as suitable, satisfying or attractive to programme recipients (people with disabilities and other potential PLA-D members) | Reported satisfaction; perceived appropriateness; intent to continue use | Direct observation; qualitative data; questionnaire |
| Demand | … is PLA-D likely to be used? | PLA-D group attendance; perceived demand; expressed interest | Monitoring data; qualitative data |
| Implementation | … can PLA-D be successfully delivered to intended participants? | Delivery of meetings; success or failure of PLA-D implementation; facilitators and barriers to implementation | Qualitative data; direct observation |
| Practicality | … can PLA-D be carried out with intended participants using existing means, resources and circumstances | Solutions implemented; positive/negative impacts on participants; cost analysis | Monitoring data; qualitative data |

PLA-D, Participatory Learning and Action for Disability.

## ETHICS AND DISSEMINATION
### Ethics and consent
Ethical approval for the study has been received from the London School of Hygiene & Tropical Medicine, the Uganda Virus Research Institute ethics committee and the Ugandan National Council for Science and Technology.

Consent to participate in the study will be sought from each participant only after a full explanation has been given, an information leaflet offered and time allowed for consideration. Signed participant consent will be obtained (including thumbprint if necessary). The right of the participant to refuse to participate without giving reasons will be respected. All participants are free to withdraw at any time from the study without giving reasons and without prejudicing further treatment or PLA-D group participation. For any participant with communication or intellectual impairments, we will seek consent from their caregiver. Carer/parent interviews will also be used for children with disabilities below the age of 18. We will seek verbal assent from children aged 8–17 years for a questionnaire to be completed on their behalf.

There is a chance that participants may become distressed or upset during the interviews when discussing stigma and discrimination. To minimise the risk of distress during interviews/questionnaires/FGDs: (1) the right of the participant to refuse to answer questions or withdraw from the study at any point will be emphasised and (2) interviewers will be trained to listen non-judgementally, provide short breaks, be aware of signs of distress, fatigue or anxiety and to address concerns appropriately and signpost to available support mechanisms.

### Referrals and reimbursement
Group facilitators will make PLA-D group participants aware of local health services available and will encourage them to attend for services if they have an expressed healthcare need. If disclosures are made in the group of safety concerns (eg, experience of violence) then the group facilitator will liaise with the local researcher for course of action.

We will provide reimbursement of 25 000 Ugandan Shillings (approximately £5.60) for participants in interviews, questionnaires, or FGDs to compensate them for their time, and in compliance with Ugandan guidance. Refreshments will be provided at group meetings.

### Dissemination plan
Research is most worthwhile if it can be used to improve policy and practice. We will develop tailored strategies for engagement and communication with our key audience to facilitate influence of policy and practice, using our strong national and international networks, including:

► Community members and people with disabilities: hosting dissemination meetings in the community and with OPDs.
► Policy and programme stakeholders in Uganda (eg, health systems actors, OPDs and NGOs): individual meetings with key organisations, production of policy and evidence briefs, relevant to Uganda.
► International policy and programme stakeholders: production of policy and evidence briefs, including through the Missing Billion Initiative, meetings with key stakeholders, contribution to WHO World Report on Disability and Health.
► Academics: peer-reviewed journal articles, presentations at conferences/seminars/webinars and use of material in teaching, including LSHTM's online teaching and online research seminars (eg, LSHTM, Uganda Research Unit).

### Potential limitations
A key limitation is that the pilot study will only be conducted in one setting. Consequently, it will only allow assessment of some domains of the Bowen feasibility framework (acceptability, demand, implementation, practicality) but not others (adaptation, integration, expansion, limited efficacy). It will therefore not be possible to make inferences on feasibility in other settings, or potentials for scale and impact. Moreover, we will include a limited number of participants in the pilot study, which will make it difficult to draw inferences for feasibility for different subgroups, such as people with different impairment types. We plan to undertake a full RCT following the feasibility study, which would allow assessment of impact and exploration of feasibility for different subgroups. There is a concern that participants will respond positively about the feasibility of the intervention because of social desirability bias. We will therefore use objective markers of feasibility (eg, group attendance), as well as subjective measures (eg, reported satisfaction), and will train the interviewers carefully about how to avoid this bias (eg, asking neutral questions).

**Author affiliations**
[1]International Centre for Evidence in Disability, Department of Population Health, London School of Hygiene and Tropical Medicine, London, UK
[2]MRC/UVRI and LSHTM Uganda Research Unit, Entebbe, Uganda
[3]Division of Physiotherapy, Department of Health and Rehabilitation Sciences, Stellenbosch University, Cape Town, South Africa
[4]Women and Children First, London, UK
[5]School of Public Health, Makerere University College of Health Sciences, Kampala, Uganda
[6]Department of Global Public Health, Karolinska Institutet, Stockholm, Sweden
[7]Amref Health Africa in Uganda, Kampala, Uganda

**Contributors** The study concept and design was conceived by HK, TS, FBM and PW. JD, MR, PK and PW informed the development of PLA-D. FBM and ASS will lead on the data collection. Analysis will be performed by HK, FBM and ASS. HK prepared the first draft of the manuscript. All authors provided edits and critiqued the manuscript for intellectual content.

**Funding** This work is supported by NIHR grant number NIHR301621. Conceptualisation of the programme was supported by the PENDA grant from the Foreign, Commonwealth and Development Office.

**Competing interests** None declared.

**Patient and public involvement** Patients and/or the public were involved in the design, or conduct, or reporting, or dissemination plans of this research. Refer to the Methods section for further details.

**Patient consent for publication** Not applicable.

**Provenance and peer review** Not commissioned; externally peer reviewed.

**ORCID iDs**
Hannah Kuper http://orcid.org/0000-0002-8952-0023
Tracey Smythe http://orcid.org/0000-0003-3408-7362

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
