## [Reviewer comments · BMJ Open]

ARTICLE DETAILS

TITLE (PROVISIONAL)	Is it feasible to implement a community-based participatory group programme to address issues of access to healthcare for people with disabilities in Luuka district Uganda? A study protocol for a mixed-methods pilot study
AUTHORS	Kuper, Hannah; Ssemata, Andrew Sentoogo; Smythe, Tracey; Drazdzewska, Joanna; Waiswa, Peter; Kagurusi, Patrick; Rosato, Mikey; Mbazzi, Femke Bannink

VERSION 1 – REVIEW

REVIEWER	Rathore, Farooq PNS Shifa Hospital
REVIEW RETURNED	07-May-2023

GENERAL COMMENTS	I would like to begin by acknowledging the importance of your study protocol in the context of disability management in the LIC/LMIC.. However, I must point out that the current manuscript is not ready for publication in its current form. Some deficiencies in the method section need to be addressed to provide a clear understanding of the study design, procedures, and limitations. I commend the authors for their efforts in conducting this interesting study. However, I strongly recommend that the manuscript be revised and expanded to address the issues mentioned above to improve its quality and readiness for publication. Please see my detailed comments below and address them in the revised version Best regards. Method Section Overall, the methods section appears to be well-designed and comprehensive, but there are a few areas for improvement: 1. Clear definition of disability: The methods section does not provide a clear definition of disability, which may lead to inconsistency in identifying participants. Providing a clear definition of disability at the outset of the study will help ensure consistency in participant selection.2. Lack of detail on sample size determination: The methods section does not provide a clear description of how the sample size was determined. It would be helpful to provide more detail on the criteria used for selecting the sample size.3. Lack of detail on recruitment: While the methods section describes how participants will be identified, it does not provide a clear description of how they will be recruited. A detailed description of the recruitment process would be helpful.
--

	4. Lack of detail on training of facilitators: While the methods section describes that facilitators will be trained; it does not provide a clear description of what the training will entail. Providing more detail on the content and length of the training would be helpful. 5. Lack of detail on potential limitations: The methods section does not provide a clear description of potential limitations of the study. It would be helpful to provide a clear description of potential limitations, including potential biases and limitations of the study design. This section was addressed earlier in the abstract but it would be appropriate to include it here for clarification. In summary, the methods section would benefit from providing more detail on the sample size determination, recruitment, training of facilitators, data management, data analysis plan, ethical considerations, and potential limitations of the study. Analysis Plan/ Dissemination of Findings The section provides a clear and comprehensive overview of the data management and analysis plan for the research study. It outlines how the interviews and focus group discussions will be recorded and transcribed, and how the data will be anonymized and stored securely to comply with the General Data Protection Regulation requirements. The section also highlights the methods that will be used for analyzing both the qualitative and quantitative data, and how the findings will be used to evaluate the feasibility of PLA-D implementation. The involvement of people with disabilities in the development, pilot-testing, and implementation of PLA-D is emphasized, which is a good example of patient and public involvement in research. Overall, the section is well-written, detailed, and transparent about the research data management and analysis plan, which is important for ensuring the credibility and reproducibility of the research findings.
--	---

REVIEWER	Cohen, Flora Washington University in St Louis
REVIEW RETURNED	21-May-2023

GENERAL COMMENTS	I appreciated reading this protocol, it is important and timely. I hope to see more work that supports community-based programming with individuals with disabilities. The protocol is interesting and well-written. The development of the PLA-D program appears to be thorough, and the setting for the implementation and feasibility testing is appropriate. However, there are a few minor revisions that would make this manuscript stronger. I have commented the suggestions in the attached file, but I will note a summarized version here as well. Abstract:  - minor grammar and spelling typos on line 6 and line 21 - I don't think the quantitative analysis would be efficacy testing, perhaps effectiveness, but you are likely testing reliability and validity of the research instruments rather than the effectiveness of the program. - if the trial is registered, please include the registration number. Methods:
---

	 - Please provide more information about the setting, including known demographic characteristics if possible. - As recommended by the journal, please provide the approximate dates that the program will start and end - There is no information about the people who will be coding and analyzing the data. Please provide information about the people who will be completing this phase of the process and their training. - provide more information in the data analysis plan, especially for the quantitative methods. I would recommend consulting with a statistician about the most appropriate approach for the intended aims.
--	--

VERSION 1 – AUTHOR RESPONSE

Reviewer: 1
Method Section

Overall, the methods section appears to be well-designed and comprehensive, but there are a few areas for improvement:

1. Clear definition of disability: The methods section does not provide a clear definition of disability, which may lead to inconsistency in identifying participants. Providing a clear definition of disability at the outset of the study will help ensure consistency in participant selection.

Response: We have included a section on definition of disability in the methods, and have specified that for the pilot study people will self identify as disabled or not.

2. Lack of detail on sample size determination: The methods section does not provide a clear description of how the sample size was determined. It would be helpful to provide more detail on the criteria used for selecting the sample size.

Response: We have clarified in the methods that the sample size was selected on the basis of 1) logistical feasibility (time/cost) and 2) our previous experience of conducting a feasibility study of a group based intervention.

3. Lack of detail on recruitment: While the methods section describes how participants will be identified, it does not provide a clear description of how they will be recruited. A detailed description of the recruitment process would be helpful.

Response: Participants will be identified and recruited through the community engagement meetings at the 5 different parishes where information about the study and groups will be provided and those interested in participation will be asked to sign up/ register their names and contact and will be contacted individually by the research team. Additional people with disabilities will be invited to attend by the focal person with disabilities at the district and people with disabilities who attended the community engagement meetings. This is now specified in the methods.

4. Lack of detail on training of facilitators: While the methods section describes that facilitators will be trained; it does not provide a clear description of what the training will entail. Providing more detail on the content and length of the training would be helpful.

Response: The facilitators, managers and supervisor will be involved in a week-long hands-on PLA-D training based on the developed manual. The training will cover aspects including understanding of disability, the PLA-D cycle, how group meetings will be conducted and management and facilitation of the groups in the community – (safeguarding, managing expectations, progress recording and reporting). This is now specified in the methods.

5. Lack of detail on potential limitations: The methods section does not provide a clear description of potential limitations of the study. It would be helpful to provide a clear description of potential limitations, including potential biases and limitations of the study design. This section was addressed earlier in the abstract but it would be appropriate to include it here for clarification.

Response: A section has been included on potential limitations

In summary, the methods section would benefit from providing more detail on the sample size determination, recruitment, training of facilitators, data management, data analysis plan, ethical considerations, and potential limitations of the study.

Response: We have attempted to address these points. Below, it states that the reviewer states that the data management and analysis plans were satisfactory, and so these have not been changed. It is not clear which clarifications in ethical considerations were required.

Analysis Plan/ Dissemination of Findings

The section provides a clear and comprehensive overview of the data management and analysis plan for the research study. It outlines how the interviews and focus group discussions will be recorded and transcribed, and how the data will be anonymized and stored securely to comply with the General Data Protection Regulation requirements.

The section also highlights the methods that will be used for analyzing both the qualitative and quantitative data, and how the findings will be used to evaluate the feasibility of PLA-D implementation. The involvement of people with disabilities in the development, pilot-testing, and implementation of PLA-D is emphasized, which is a good example of patient and public involvement in research.

Overall, the section is well-written, detailed, and transparent about the research data management and analysis plan, which is important for ensuring the credibility and reproducibility of the research findings.

Response: Thank you for these comments. No actions needed.

Reviewer: 2

1. The protocol is interesting and well-written. The development of the PLA-D program appears to be thorough, and the setting for the implementation and feasibility testing is appropriate. However, there are a few minor revisions that would make this manuscript stronger. I have commented the suggestions in the attached file, but I will note a summarized version here as well.

Response: Thank you for these comments. We have attempted to address all points raised.

Abstract:

- minor grammar and spelling typos on line 6 and line 21
This change has been made.

- I don't think the quantitative analysis would be efficacy testing, perhaps effectiveness, but you are likely testing reliability and validity of the research instruments rather than the effectiveness of the program.

We have changed this to a focus on developing and refining evaluation tools. Throughout, we have removed mention of limited efficacy testing.

- if the trial is registered, please include the registration number.

Response: The trial will only be conducted in 2024 onwards, and has not yet been registered.

Methods:

- Please provide more information about the setting, including known demographic characteristics if possible.

Response: More information has been provided.

- As recommended by the journal, please provide the approximate dates that the program will start and end

Response: These dates have been provided

- There is no information about the people who will be coding and analyzing the data. Please provide information about the people who will be completing this phase of the process and their training.

Response: This information is now provided.

- provide more information in the data analysis plan, especially for the quantitative methods. I would recommend consulting with a statistician about the most appropriate approach for the intended aims.

Response: More information is now provided, formulated by an epidemiologist with trial experience (HK).

VERSION 2 – REVIEW

REVIEWER	Cohen, Flora Washington University in St Louis
REVIEW RETURNED	19-Jul-2023
GENERAL COMMENTS	Excellent work, I look forward to seeing it in print!

VERSION 2 – AUTHOR RESPONSE